# A Community-Based Participatory Framework to Co-Develop Patient Education Materials (PEMs) for Rare Diseases: A Model Transferable across Diseases

**DOI:** 10.3390/ijerph20020968

**Published:** 2023-01-05

**Authors:** Marta Falcão, Mariateresa Allocca, Ana Sofia Rodrigues, Pedro Granjo, Rita Francisco, Carlota Pascoal, Maria Grazia Rossi, Dorinda Marques-da-Silva, Salvador C. M. Magrinho, Jaak Jaeken, Larisa Aragon Castro, Cláudia de Freitas, Paula A. Videira, Luísa de Andrés-Aguayo, Vanessa dos Reis Ferreira

**Affiliations:** 1Institute of Hygiene and Tropical Medicine (IHMT), NOVA University Lisbon, 1349-008 Lisbon, Portugal; 2CDG & Allies—Professionals and Patient Associations International Network (CDG & Allies—PPAIN), Department of Life Sciences, School of Science and Technology, NOVA University of Lisbon, 2819-516 Caparica, Portugal; 3Institute of Biomolecular Chemistry, National Research Council of Italy, 80078 Pozzuoli, Italy; 4Department of Environmental, Biological and Pharmaceutical Sciences and Technologies, University of Campania “Luigi Vanvitelli”, 81100 Caserta, Italy; 5UCIBIO, Department of Life Sciences, NOVA School of Science and Technology, NOVA University of Lisbon, 2819-516 Caparica, Portugal; 6Associate Laboratory i4HB—Institute for Health and Bioeconomy, School of Science and Technology, NOVA University Lisbon, 2819-516 Caparica, Portugal; 7Portuguese Association for Congenital Disorders of Glycosylation (CDG), Department of Life Sciences, School of Science and Technology, NOVA University Lisbon, 2819-516 Caparica, Portugal; 8IFILNOVA—Institute of Philosophy—Faculty of Social Sciences and Humanities, NOVA University of Lisbon, 1069-061 Lisbon, Portugal; 9ALiCE—Associate Laboratory in Chemical Engineering, Faculty of Engineering, University of Porto, Rua Dr. Roberto Frias, 4200-465 Porto, Portugal; 10LAQV (Associate Lab for Green Chemistry)—Chemistry Department, NOVA School of Science and Technology, NOVA University of Lisbon, 2819-516 Caparica, Portugal; 11Centre of Metabolic Diseases, Department of Pediatrics, KU Leuven, 3000 Leuven, Belgium; 12Swiss Personalized Health Network (SPHN), 3001 Bern, Switzerland; 13EUPATI—European Patient’s Academy on Therapeutic Innovation, 3008 Bern, Switzerland; 14SNSF Swiss National Science Foundation, 3001 Bern, Switzerland; 15Laboratório para a Investigação Integrativa e Translacional em Saúde Populacional (ITR), 4050-600 Porto, Portugal; 16EPI Unit—Instituto de Saúde Pública, Universidade do Porto, 4050-600 Porto, Portugal; 17Departamento de Ciências da Saúde Pública e Forenses e Educação Médica, Faculdade de Medicina, Universidade do Porto, 4200-319 Porto, Portugal; 18Center for Genomic Regulation (CRG), Barcelona Biomedical Research Park (PRBB), 08003 Barcelona, Spain

**Keywords:** patient education material (PEM), rare diseases, community-based participatory research, public and patient involvement (PPI), health literacy, patient empowerment, people-centric, plain-language, congenital disorders of glycosylation (CDG)

## Abstract

At least 50% of chronic disease patients don’t follow their care plans, leading to lower health outcomes and higher medical costs. Providing Patient Education Materials (PEMs) to individuals living with a disease can help to overcome these problems. PEMs are especially beneficial for people suffering from multisystemic and underrecognized diseases, such as rare diseases. Congenital disorders of glycosylation (CDG) are ultra-rare diseases, where a need was identified for PEMs in plain language that can clearly explain complex information. Community involvement in the design of PEMs is extremely important for diseases whose needs are underserved, such as rare diseases; however, attempts to involve lay and professional stakeholders are lacking. This paper presents a community-based participatory framework to co-create PEMs for CDG, that is transferable to other diseases. A literature review and questionnaire were performed, and only four articles describing the development of PEMS for rare diseases have been found, which demonstrates a lack of standardized approaches. The framework and PEMs were co-developed with CDG families and will be crucial in increasing health literacy and empowering families. We will close a gap in the creation of PEMs for CDG by delivering these resources in lay language in several languages.

## 1. Background

The definition of health literacy emerged in the health sector with a narrow focus on reading and writing health materials, but it has been broadened to knowledge reflecting a more holistic health promotion approach, shifting away from an individual skill towards a community skill (Appendix A) [1,2,3,4,5,6,7,8,9,10,11,12,13,14,15,16,17,18,19,20,21,22,23,24,25,26,27,28].

Health literacy is people’s ability to obtain, comprehend, and apply health information to make decisions with their healthcare provider about illness prevention and health promotion. This should lead to maintaining or improving quality of life [27].

Mancuso and colleagues have underlined that health literacy is a lifelong learning process with capacity, comprehension, and communication as central attributes, that affects not only individuals but also society (Appendix A) [19].

Several health literacy models have been developed in recent years [29]. The European Health Literacy Consortium for the European Health Literacy Survey has created an Integrated Model of Health Literacy (IMHL). The IMHL represents the capabilities related to the process of retrieving, understanding, accessing, and applying information [27]. When these four competencies referring to health information processing are combined with the three levels of domains named healthcare, disease prevention, and health promotion, a matrix of 12 subdimensions of health literacy is produced (Appendix A) [27]. The model can be used to develop and validate measurement tools, as well as to help create health literacy strengthening interventions [27]. However, only a few studies have developed measurement tools based on the IMHL [30,31].

Based on current data, establishing health literacy programs and monitoring methods is a priority. According to the Centers for Disease Control and Prevention (CDC) nearly nine out of ten adults face challenges in understanding how to proceed when their doctor, nurse, health administrator, pharmacist, or other healthcare professional presents complicated, verbal or written, health information [32]. Low health literacy is associated with poor and limited knowledge about medical conditions [33], less use of preventive health services [34,35], non-adherence to treatment plans and medical regimens [33], poor patient self-care [36,37,38], poorer outcomes [39], high healthcare costs, and increased risk of hospitalization and mortality [40].

Limited health literacy harms individuals’ health and well-being and costs money to the healthcare system, impacting the country’s financial health. Improving health literacy could prevent nearly 1 million hospital visits and save over $25 billion a year [32]. Low health literacy is a factor contributing to healthcare disparities [41] and social determinants of health [42]. Healthcare providers should consider health literacy a vital component when caring for people of any age, culture, education, or socioeconomic status [43].

To increase health literacy levels, different strategies and tools [44] have been developed, such as Patient Education Materials (PEMs), and various actions at the political level have been encouraged and/or implemented [45,46,47,48]. However, health literacy is not just the responsibility of individuals, policymakers, or healthcare professionals, it crosses multiple boundaries, professions, and sectors [30,49].

PEMs are a knowledge translation intervention for the dissemination of reliable clinical information (such as clinical practice guidelines or journal articles) [50]. They address prevalent health issues and commonly requested health topics. Materials should be adapted to the population’s reading and comprehension levels as well as to its cultural and ethnic diversity. Thus, PEMs are a potential means to explain important topics and are a vehicle to increase patient–doctor dialogue. Well-formulated and well-presented healthcare information increases health awareness, encourages self-care, and improves the effectiveness of clinical care [50].

More PEMs are being developed, although they don’t always correspond to diverse diseases and reading levels [45,47]. The PEMs assessment tool (PEMAT) was created and serves as a guide to help determine whether patients will be able to understand information and act accordingly [45].

People living with rare diseases and their families are among the most vulnerable population groups in terms of health literacy. With much less information about rare diseases, it is harder for parents, and even doctors, to get accurate information about the diagnosis, care, treatment, and family support [51]. The development of PEMs for rare diseases is hampered by difficulties accessing reliable, easy-to-understand information about the conditions, clinical practices, and patients’ needs. These constraints contribute to the scarcity of PEMs in the rare diseases field, which increases isolation and powerlessness, lowering the quality of life [51].

Congenital disorders of glycosylation (CDG) are rare metabolic diseases caused by genetic biosynthetic defects in glycosylation pathways. They are primarily complex, multisystem diseases. The CDG & Allies international network, in collaboration with the CDG community, identified a community need, revealing a widespread desire among families and professionals for lay language resources with clear and simpler ideas [52,53,54,55].

A people-centric framework or community-based framework is an important aspect in the development of inclusive and relevant PEMs. By involving not only professionals but also families in the development of new PEMs, it is possible to create resources directed to people’s needs, values, and preferences [53]. This methodology refers to the involvement of individuals living with CDG and/or families as active equal partners and whose insights and beliefs are continuously incorporated throughout the PEMs co-creation process [56,57].

However, there is a lack of standardized approaches to create educational materials for rare diseases like CDG and common diseases. This is supported by the conducted literature review and questionnaire [56]. Therefore, we developed a people-centric framework with the goal of creating digital and printed PEMs that will empower families by increasing their health literacy. From this framework, it was possible to create several resources in lay language translated into different languages in order to fill the lack of dispersion of educational materials available for the CDG community [57,58].

## 2. Materials and Methods

### 2.1. Assessing Families’ Information Needs: CDG Journey Mapping Questionnaire

The CDG Journey Mapping electronic questionnaire (e-questionnaire) has been performed as described previously by Francisco et al. [59]. This questionnaire seeks to understand the journey of the person who lives with CDG, from diagnosis to informational needs and healthcare/social support measures. This questionnaire has two versions but, in this study, we use exclusively the e-questionnaire version for CDG patients, family members, and caregivers (Figure 1) [59]. The finished CDG families’ version of the questionnaire had a total of 96 questions in different formats, namely multiple-choice, matrix, and open-ended questions. IP addresses were not saved to ensure respondents’ anonymity, and the logic functionality was used to guide respondents.

The ethics committee of the Faculty of Psychology, University of Lisbon, granted ethical approval for this study (reference 1.15/07/2021-22). All participants consented electronically.

From this questionnaire it was possible to extract information from topics related to the journey to get a diagnosis, as well as informational needs (from the time of diagnosis until the moment they answered the questionnaire). However, only pertinent questions to assess the CDG community’s needs for PEMs were chosen for this study.

### 2.2. Literature Search on Frameworks for the Development of PEMs

A literature review was conducted to understand the state of the art regarding the existence of established frameworks for the creation of PEMs in rare diseases. With that aim, an automated python search tool was used with a list of keywords using double and triple combinations of terms related to rare diseases, CDG, and PEMs to identify articles through the Medline database using PubMed as the search engine (Appendix A), as described by Brasil et al. [60]. Selection criteria were chosen to screen the extracted articles in order to focus on relevant literature for this study, obtain the most recently reported work, and eliminate irrelevant and unrelated work (Table 1). Following the aforementioned criteria, exclusion and inclusion criteria were carried out, as well as a comparison of the number of articles available describing the development of patient education materials for rare diseases versus other diseases.

### 2.3. Community-Based Framework to Develop Patient Education Materials

#### Set Up of a Community-Based Framework

To try to answer to the previously assessed needs a community-based framework was developed for Patient Education Materials.

CDG and Allies—Professionals and Patient Associations International Network (PPAIN) and the Portuguese Association for Congenital Disorders of Glycosylation (APCDG) offered educational resources (Appendix A). Each audience has a different learning style; these materials are designed in order to adapt to the needs and preferences of different individuals. Once a style is identified, several best formats for patient education materials should be made available (Appendix A). 

A community-based framework was generated to guide the creation of PEMs, by involving not only professionals but also the families in the development of new resources. This framework produces two different documents: infographics and summaries. These documents summarize highly complex information about a specific CDG in a concise and clear manner.

Figure 2 shows how PEMs were co-created in 8 phases.

Identification of the informational need

The co-creation process begins when a family or a medical professional requests information about signs and symptoms, prognosis, and/or clinical management of a specific CDG through different multi-channels (e.g., World CDG Organization Website, WhatsApp).

2.Assembly of the team

Based on the requested information, a team is assembled consisting of: (1) a researcher and/or project manager in charge of managing the team, recruiting the team, and transferring the information among the different members, (2) a researcher or a student from life sciences responsible for researching and compiling the published information, (3) a researcher responsible for an initial revision, (4) a medical/scientific board to assess scientific and medical language, (5) a family living with that CDG to ensure understandability and provide their experience, if available, (6) a set of members from the CDG & Allies, to do a final revision, and (7) translators to Portuguese, Spanish, and Italian. 

A total of 8 translators, 11 developers, 6 health professionals, and 19 CDG families worldwide (14 different nationalities) were involved in the co-creation of PEMs tailored to the needs of the community. These partners’ roles range from creators to reviewers (health professionals and families) and, finally, translators for various languages (Table 2).

3.Scientific article identification, and data analysis and extraction

The identification of articles on different databases (OMIM, PubMed, GeneCards, MalaCards, LOVD3) is performed using a set of keywords (gene name, gene name-CDG); this allows an in-depth search of the existing bibliography about a CDG. 

Identified articles are downloaded and an Excel document is filled with the main information: title of the paper, year, author, number of patients per report, country of origin of each patient (if available), types of mutations (variants), inheritance mode, diagnostic tests, clinical manifestations, management strategies, and treatments. 

Information is extracted to write a summary and the content for an infographic. The first one is more extensive and detailed, and is directed to healthcare professionals and researchers. The second one uses simpler language and has more succinct and clear ideas, as it is directed to the general public, like families. 

4.Revision process

When the content for the infographic and summary are ready, a comprehensive revision process ensures that the final material is scientifically and medically accurate, and easy to understand for the general public.

Researchers from CDG & Allies review the summary and the content for the infographic, grammar, and sentence structure.

The next step is to send this first draft to the medical/scientific board for revision. The reviewer should ensure that the correct medical terminology has been used throughout the content for the infographic and summary. The second draft of the content for the infographic is sent to the family (if available) living with this type of CDG for a plain-language revision. Families must ensure that content is straightforward and understandable.

The content for the infographic and summary is submitted to a researcher who never saw the first draft for the last validation.

5.Graphic design

A graphic designer originates an attractive and informative infographic using the previously validated content, and photographs from a person living with CDG. These photos are requested from the family participating in the revision process. When no family is available, an image from an article is used as inspiration. The core team validates the infographic’s design and content.

6.Translation of the English infographics

Afterward translations are developed in Portuguese, Spanish, and Italian. The translations are reviewed by a health expert with fluency in the language translated.

7.Communication campaigns

The infographics are used for a multi-channel educational and awareness campaign including social media, worldcdg.org, World CDG Magazine, CDG tailored events such as World Conference on CDG, and so forth.

## 3. Results

### 3.1. CDG Community Information Needs

A total of 174 CDG families and caregivers completed the e-questionnaire. Most CDG family respondents (86.6%) were relatives or family caregivers. The participants were spread globally, although most of them are located in the European continent (44.2%). In addition, 67.8% of the family members had at least a bachelor’s degree (Figure 3). In this study, which focused on informational accessibility and PEMs, 3 questions were chosen out of 96. 

The majority of the respondents took 1 to 3 years (28.74%) to get a definitive diagnosis, followed by 7 to 12 months (18.4%), and 3 to 6 months (14.9%). At the time of diagnosis, most of the respondents (70.6%) stated a lack of understanding of the healthcare professional’s given information (58.0%—Partially Understood, 12.6%—Did not Understand), with a minor percentage of respondents having a complete understanding of the given information. Lack of information in their native language (32.8%) and difficulties accessing information in their country (29.9%) are the biggest challenges to accessing appropriate CDG information for families; 67.2% said specialized information for rare diseases is the largest challenge (Figure 4).

### 3.2. Existence of Frameworks for PEMs

We compared the number of articles available describing the development of patient education materials for rare diseases to other diseases in order to understand the state of the art in the development of PEMs and the creation of frameworks for PEMs (Figure 5). Compared to common diseases, rare diseases had fewer publications (in total 3).

### 3.3. Implementation of the Created Community-Centric Framework in CDG

The applied methodology has allowed the development of several infographics dedicated to different CDG types (Table 3), such as the one shown in Figure 6. By November 2022, 32 infographics had been created. Furthermore, four infographics are undergoing different phases of development (Table 3). 

The 32 PEMs were and are being translated with accurate cultural adaptations by a group of native speakers composed mostly of health professionals and researchers from the CDG & Allies—PPAIN network team. Infographics are available in three distinct languages: Spanish, Portuguese, and Italian. Currently, all existing documents are available on the World CDG organization platform (https://worldcdg.org/ (accessed on 22 November 2022)). 

## 4. Discussion

It is critical to create PEMs with a people-centric framework in order to improve community and clinical outcomes. In this study, we established a framework for PEMs and highlighted the CDG community’s informational need and the scarcity of pre-existing frameworks to develop PEMs for rare diseases.

Almost half of the respondents had to wait 1–3 years for a definitive diagnosis. This demonstrates a lack of awareness among healthcare professionals due to CDG being an ultra-rare disease associated with a high level of complexity and the plethora of signs, symptoms, and systems affected. This contributes to misdiagnosis and diagnostic delay [61,62]. It is also clear that patients and families have difficulty understanding the healthcare professionals at the time of diagnosis (70.6%). This can be explained by the previously mentioned arguments emphasizing the rarity and complexity of the disease. Therefore, due to their lack of familiarity with this group of ultra-rare diseases, the physician in charge of providing the diagnosis is unlikely to be able to translate the information in a way that the families can understand. Therefore, both results emphasize the need for the development of PEMs in lay-language which empowers both families and professionals. This is further supported by the fact that one of the major barriers is the lack of tailored specialized information on rare diseases, notably in easy-to-understand language, which is linked to the absence of research and available resources [63]. However, this information should be accessible to everyone in the vastest set of languages since families pointed out the lack of information in their native language as a barrier to accurate information (32.8%).

The creation of these educational materials should consider both families and professionals, for example by creating a framework that translates scientific jargon into lay language for clinical management guidelines that are essential resources for both healthcare professionals and families [59].

The conducted literature review showed that common diseases had more publications (*n* = 34) dedicated to frameworks that create PEMs, when compared with rare diseases (*n* = 3) [51,64,65]. This reveals that there is an overall lack of user-friendly information across all diseases, and it is an area with a significant growth potential in the upcoming years. In terms of information, regarding the three articles focusing on developing PEMs to improve family’s health knowledge, decision-making skills, and clinical outcomes, they all described a patient-centric framework for rare diseases, emphasizing the importance of the involvement of the community [51,64,65].

The delay in creating these resources denies families access to high quality resources, limiting their understanding of their rare disease and increasing their isolation and helplessness. To address this, some companies and research groups have devised several tools, based on artificial intelligence and advanced software, to quickly and cheaply develop materials in lay language. Additional research was carried out to assess the availability of these tools but there are no reports published yet (Appendix A). However, through desk research SumMed was identified. This tool was created to assist patients and families navigate medical information and make educated decisions. SumMed is an application where people put their medical reports or some scientific article, and then the user receives an easy-to-understand summary after a few seconds [66]. The patient also receives information about the most credible sources regarding their problem. This resource is currently available in 60 languages. This tool will democratize medical information, make it accessible and understandable to patients, informal caregivers, and families. Ultimately, it can enable the overall creation of PEMs [66].

This article tries to overcome an identified unmet need across all diseases by describing a community-based participatory framework to co-develop educational resources for CDG in multiple languages. In total 32 infographics have been completed, 29 have been translated into Spanish, 30 into Italian, and 23 into Portuguese. Soon, more infographics will be developed and translated using this framework. Many other diseases can use and produce PEMs by replicating these community-based participatory frameworks. 

### 4.1. Study Strengths and Limitations

The creation of a standardized method for producing PEMs for CDG in lay language is an innovative initiative. It requires a wide range of resources, including collaboration, funding, and countless hours of work. The dispersion of qualitative information, the need to compile it in simple but scientifically and medically correct language, and the demand for translations in several languages all contribute to the lengthy process of developing PEMs. Our framework relies on community involvement. The CDG community is willing to participate and co-create PEMs [53,67].

The development of PEMs as an approach to increasing health literacy should be an integrated model, involving a diverse range of stakeholders, including health professionals, non-governmental organizations, scientific researchers, pharmaceutical companies, and last but not least families. The various stakeholders will benefit among other things from faster diagnosis, improved understanding of the condition and improved communication between families and healthcare providers among others.

The designed framework’s adaptability and flexibility enable new adaptations for other diseases, especially for rare diseases where there are limited plain-language PEMs. PEM development for rare diseases has various hurdles, including a lack of researchers, health professionals, and families involved in the co-creation of PEMs, as well as limited funding. These challenges highlight the need for methodologies that standardize the creation of these resources while ensuring applicability to other rare disease communities [43].

### 4.2. Future Perspectives

Several resources were created based on the developed methodology and are currently available on the World CDG organization platform (https://worldcdg.org (accessed on 22 November 2022)), with a special section for plain-language PEMs. Since the CDG community is global and multilingual, increasing the number of languages into which the PEM resources are translated is a top priority. This is consistent with the findings of the CDG Journey Mapping questionnaire [52,53,59]. 

We will use the Patient Education Materials Assessment Tool, which is a systematic method for evaluating the understandability and feasibility of the materials [45], to ensure that the resources we create are clear and easy to understand. When validating infographics, this type of tool would allow professionals to eliminate variables such as the family’s pre-existing literacy level.

We also intend to assess the level of Health Literacy among the CDG community, using the HLS-EU-Q47, a 47-item multidimensional and comprehensive questionnaire that measures the health literacy level of the general population. This questionnaire was developed as part of the Consortium of the European Health Literacy Project (HLS-EU 2009–2012). It became the first comparative European Health Literacy Survey (HLS-EU) in 2011 [68,69,70]. 

Lastly, a modernization of the current format PEMs is also being considered, such as the development of podcasts and videos in order to ensure more diverse multi-channel educational campaigns.

## 5. Conclusions

PEMs empower families by allowing them to make decisions alongside their health professionals. We established a replicable health literacy community-based participatory framework to co-create PEMs. This concept could help other rare disease communities.

## Figures and Tables

**Figure 1 ijerph-20-00968-f001:**
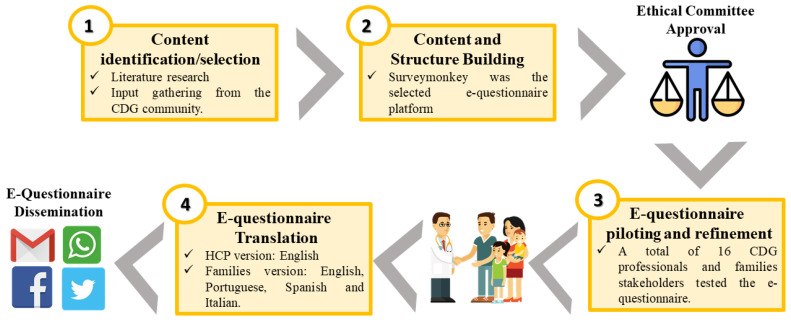
Schematic of the workflow that was used to develop the CDG Journey Mapping Questionnaire.

**Figure 2 ijerph-20-00968-f002:**
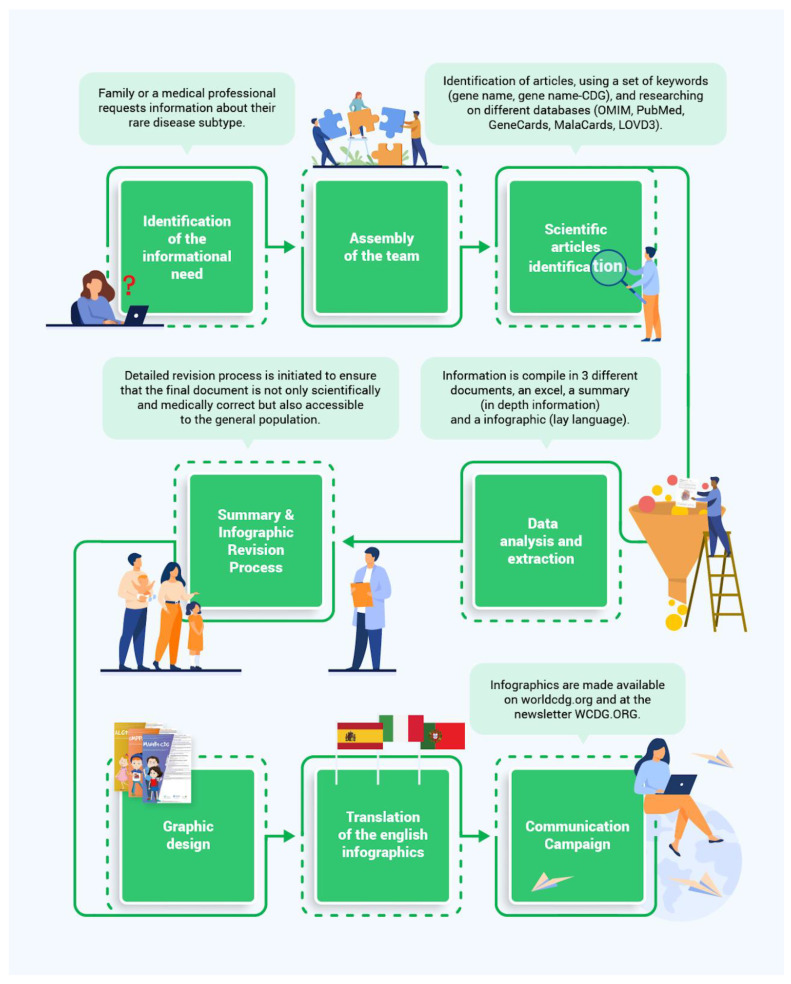
Community-based framework developed to guide the creation of PEMs.

**Figure 3 ijerph-20-00968-f003:**
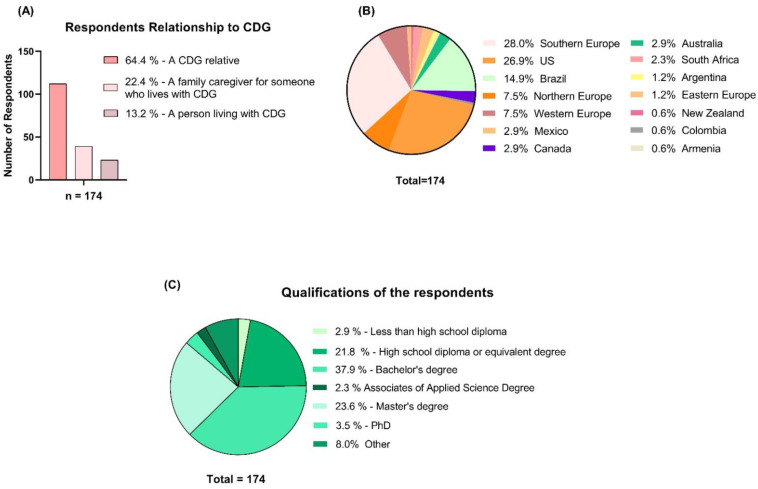
Socio-demographical characterization of the CDG Journey Mapping e-questionnaire respondents included in this study (CDG family members and caregivers). (**A**) Respondents’ relationship with CDG. (**B**) Geographical distribution of the respondents (per country of residence). (**C**) Family and caregivers’ academic qualifications.

**Figure 4 ijerph-20-00968-f004:**
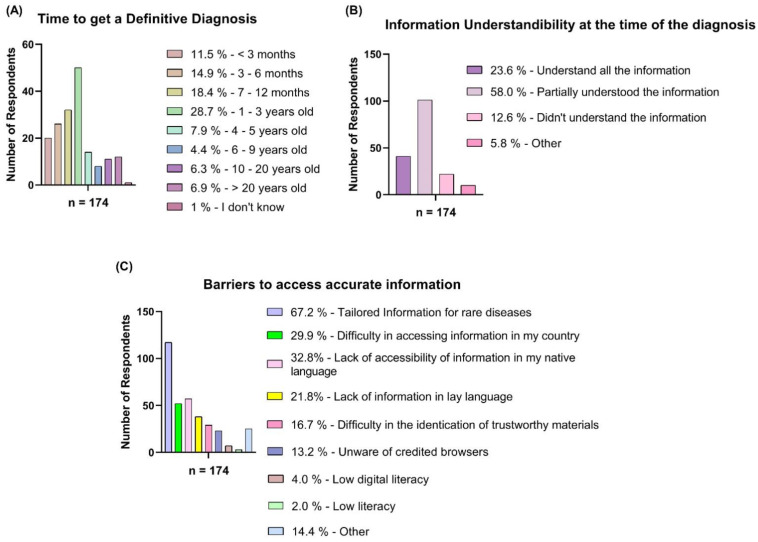
Overview of the answers submitted by the respondents included in this study (CDG family members and caregivers). (**A**) Family members and caregivers’ percentage response to the question ‘’Time to get definitive diagnosis’’. (**B**) Family members and caregivers’ percentage response to the question ‘’Time of diagnosis information understandability’’. (**C**) Family members and caregivers’ percentage response to the question ‘’The main barrier to access accurate information’’.

**Figure 5 ijerph-20-00968-f005:**
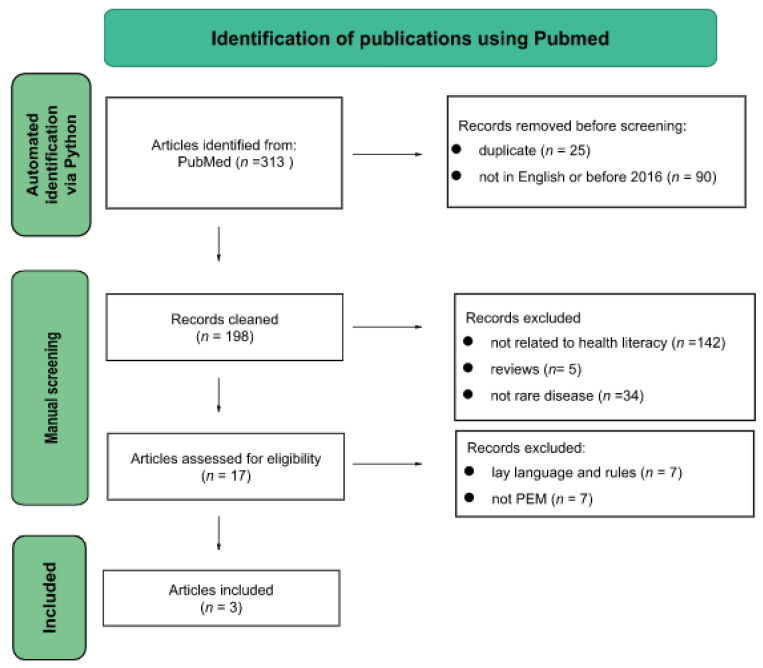
Flow diagram for the literature search which included searches of databases. Only articles related to PEMs developed for rare diseases were included. The PRISMA flow diagram was adapted from the study by Page et al. (Page, M.J.; McKenzie, J.E.; Bossuyt, P.M.; Boutron, I.; Hoffmann, T.C.; Mulrow, C.D.; Shamseer, L.; Tetzlaff, J.M.; Akl, E.A.; Brennan, S.E.; et al. The PRISMA 2020 statement: An updated guideline for reporting systematic reviews. PLoS Med. 2021, 18, e1003583.)

**Figure 6 ijerph-20-00968-f006:**
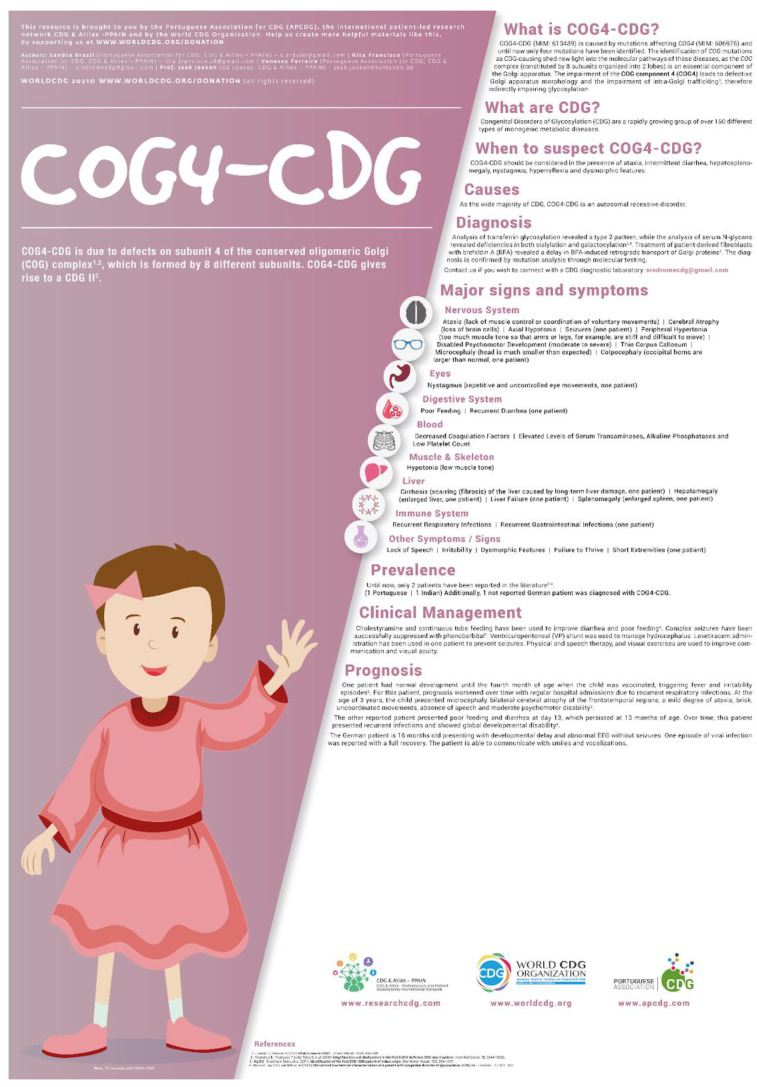
Example of an infographic dedicated to COG4 -CDG. To consult the overall infographics already available check at https://worldcdg.org/ (accessed on 22 November 2022).

**Table 1 ijerph-20-00968-t001:** Selection criteria used for the inclusion of articles of interest for this study.

Inclusion Criteria	Exclusion Criteria
Published after 2016	Health literacy papers mainly focusing on patient clinical management
Written in English	Reviews, open articles and similar articles, with the exception of those used for contextualization
Orphanet classification is used, and only rare diseases with orphacodes are accepted	
PEMs specific to a disease	
Frameworks related to the creation of the content from PEMs	

**Table 2 ijerph-20-00968-t002:** Professionals and families committee, divided by nationality and specific task. Translators: healthcare professionals, researchers and students from life sciences; developers of the PEMs: researchers and natural science students; revisors and validators: healthcare professionals and families.

Nationalities	Translators	Developers of the PEMs	Revisors and Validators	
			Medical Professionals	Families
Portuguese	2	8		1
Spanish	3	2	1	2
Italian	2			1
Italian/Swiss		1		
Belgian			1	
American			4	6
Brazilian				2
Canadian				3
Australian				1
Mexican	1			
Turkish				1
British				1
German				1
Swedish				1
Total	8	11	6	19

**Table 3 ijerph-20-00968-t003:** Finished CDG infographics by different types of glycosylation.

Types of Glycosylation	Disease Designation(s)	Final Infographics per Language
English	Portuguese	Italian	Spanish
N-linked CDG	ALG1-CDG		-
ALG2-CDG		-	
ALG3-CDG	
ALG6-CDG	
ALG8-CDG		-
ALG9-CDG	
ALG11-CDG	
ALG12-CDG	
FUT8-CDG	
MAN1B1-CDG (Mental retardation 15)	
RFT1-CDG	
MOGS-CDG	
MPI-CDG *	
PMM2-CDG	
SSR3-CDG	
SSR4-CDG		-	
GPI-biosynthesis defects	PIGA-CDG (Multiple congenital anomalies-hypotonia-seizures syndrome 2)		-	
PIGG-CDG (Mental retardation 53)	
PIGN-CDG (Multiple congenital anomalies-hypotonia-seizures syndrome 1 (milder phenotype)	
Disorders of multiple glycosylation pathways	COG4-CDG	
COG5-CDG		-		
COG6-CDG	
DOLK-CDG	
DPM1-CDG		-	
DPM2-CDG		-	
GMPPA-CDG (Alacrima, achalasia, and mental retardation syndrome)	
MPDU1-CDG		-	
NANS-CDG (Spondyloepimetaphyseal dysplasia, Camera-Genevieve type)	
PGM1-CDG	
SLC35A2-CDG (epileptic encephalopathy) (high incidence of de novo variants)	
SLC39A8-CDG		-	
O-linked CDG	B4GALT7-CDG (Ehlers–Danlos syndrome, spondylodysplastic type, 1)		-	
Total		32	23	30	29

* MPI is also translated into French, Dutch, German, and Bulgarian. 

: Finished infographics.

## Data Availability

All data generated or analysed during this study are included in this published article (and its Appendix A).

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
