# Peer review of "A Community-Based Participatory Framework to Co-Develop Patient Education Materials (PEMs) for Rare Diseases: A Model Transferable across Diseases"

_ijerph, 2023, doi:10.3390/ijerph20020968_

Round 1

Reviewer 1 Report

Dear authors:
Thank you for this interesting work.
In the summary, you explained the objectives of your study and the need to produce PEMs for patients with CDG. In the introduction you started directly with the definition of health literacy up to line 37, only then did you relate health literacy to the subject of your article. It would have been more didactic if you started with the PEMs and then made the link with health as on line 95.
The methodology is very well done
You notice that "Participants were spread all over the world, underlining the international character of the CDG community" but the majority of your participants are European and the translation was only done for three languages
Line 292: a small error in the sentence "The infographics are available in 3 distinct languages: Italian, Portuguese and Italian". Italian produced twice.

Author Response

“In the introduction you started directly with the definition of health literacy up to line 37, only then did you relate health literacy to the subject of your article. It would have been more didactic if you started with the PEMs and then made the link with health as on line 95.”

The authors appreciate the feedback on the organisation of the introduction, but would like to keep the structure as is. The authors' intention with this structure was to lead the reader from a broader concept of health literacy to the concept of PEMs. Furthermore, the creation of PEMs is done in response to the need to improve population literacy, so we believe the previous structure is appropriate.

 “You notice that "Participants were spread all over the world, underlining the international character of the CDG community" but the majority of your participants are European and the translation was only done for three languages.”

The authors proceeded to change the sentence in order to demonstrate the idea more clearly and correctly.

“Line 292: a small error in the sentence "The infographics are available in 3 distinct languages: Italian, Portuguese and Italian". Italian produced twice.”

The authors are grateful for the detection of the error and proceeded to correct it.

Reviewer 2 Report

The article covers a valuable topic that is still poorly addressed by the literature. The mixed method utilized (literature and questionnaire) is also appropriate. The images included in the manuscript are also very well-designed. The discussion could be improved using previous literature.

Author Response

“The article covers a valuable topic that is still poorly addressed by the literature. The mixed method utilized (literature and questionnaire) is also appropriate. The images included in the manuscript are also very well-designed. The discussion could be improved using previous literature.”

The comments given by the reviewer were taken into account and the discussion was improved accordingly.

Reviewer 3 Report

As stated in the paper, the goal of this work was creating digital and printed Patient Educational Materials (PEM) to empower families by increasing their health literacy. In that aim, this work shows a procedure to create accessible information material for patients with rare diseases, since it is very scarce in the literature. In addition to offering a valid and reliable standard procedure that can be applied to other pathologies. As inferred from the text, the need for this work resulted from a review of the literature, finding only 3 articles that dealt with PEMs frameworks for rare diseases. To design infographics in colloquial language, the collaboration of families and professionals was sought, and several steps were followed. Among them: A questionnaire was sent to families with a rare disease that included 96 questions, of which 3 were selected, presumably to direct the infographic, but this is not clear in the text. Searches were made in databases to select information for the PEMs; however, it is not clear if it was to obtain information to be able to answer the 3 selected 3 or the 96 questions of the questionnaire. Finally, after a synthesis of the information and passing different filters a clear and attractive product resulted.  The objective of this work is laudable and very necessary, and the procedure the authors had followed to obtain the infographics is rigorous and clear cut. However, the presentation of the information and its organization in the text results confusing at times.  For example:

Introduction (lines 73-75) Figure 1 does not show the Integrated Model of Health Literacy (IMHL) but the Schematic of the workflow that was used to develop the CDG Journey Mapping Question

Materials and Method:  The infographic that is used as an example includes information of all kinds:  on the pathology, causes, diagnoses, treatment... and does not address, as I understood, the relevant questions, (in section 3.1.) "Time to get definitive diagnosis' ', ''Time of diagnosis information ' ''The main barrier to access accurate information''. In fact, I do not understand the purpose of selecting those questions out of the 96. Please, justify the relevance of the selected questions for the PEM result.

The Results section includes a description of the procedure followed for a systematic review of the literature on PEM frameworks for rare diseases, I presume it should be in the Procedures section.

In the Discussion section reference is made to Table 4 but it has not been included.

Authors may see other comments within the text

Author Response

Introduction (lines 73-75) Figure 1 does not show the Integrated Model of Health Literacy (IMHL) but the Schematic of the workflow that was used to develop the CDG Journey Mapping Question 

The figure referred by the authors of the study on line 75 is located in the supplementary materials provided. 

Materials and Method:  The infographic that is used as an example includes information of all kinds:  on the pathology, causes, diagnoses, treatment... and does not address, as I understood, the relevant questions, (in section 3.1.) "Time to get definitive diagnosis' ', ''Time of diagnosis information ' ''The main barrier to access accurate information''. In fact, I do not understand the purpose of selecting those questions out of the 96. Please, justify the relevance of the selected questions for the PEM result.

The provided infographic was created in accordance with the framework referred to in the materials and methods, with the goal of responding to the information referred to by the CDG community in the CDG Journey Mapping Electronic (e-) Questionnaire, which is a big gap identified by our research group. The infographics were created to try to answer and empower families to common questions usually raised post diagnosis (e.g? What is CDG; Causes; Signs and Symptoms; Clinical management and prognosis.). The CDG Journey Mapping questions chosen for this article aimed to characterized the socio geographical distribution of the respondents (figure 3) and then the community's need for PEMs (figure 4).  To clarify, the authors added the reason for selecting the questions.

The Results section includes a description of the procedure followed for a systematic review of the literature on PEM frameworks for rare diseases, I presume it should be in the Procedures section. 

The authors took into account the comment and tried to reformulate the text. 

In the Discussion section reference is made to Table 4 but it has not been included.

Table 4 mentioned by the authors in the discussion is located in the supplementary materials provided.